# Long-Term Combined Training in Idiopathic Pulmonary Fibrosis: A Case Study

**DOI:** 10.3390/ijerph17145091

**Published:** 2020-07-15

**Authors:** José Naranjo-Orellana, Alfredo Santalla

**Affiliations:** 1Department of Sport and Computing, Pablo de Olavide University, 41013 Sevilla, Spain; asanher@upo.es; 2Instituto de Investigación Hospital 12 de Octubre (i+12), 28041 Madrid, Spain

**Keywords:** inspiratory muscle training, resistance training, idiopathic pulmonary fibrosis

## Abstract

A supervised combined training program was applied to a sedentary 56-year-old man with idiopathic pulmonary fibrosis (IPF) along three years, until lung transplantation. It included: (a) aerobic continuous (CT) and interval training (IT), (b) high load resistance training (RT) and (c) inspiratory muscle training (IMT). IT and IMT were applied for two years, while CT and RT could be maintained until transplantation using supplemental oxygen. Maximal inspiratory pressure (MIP) kept above 180 cm H_2_O and forced vital capacity (FVC) remained stable until lung transplantation. Peak oxygen uptake VO_2_ increased during 1.5 years before its decline, staying above the poor prognosis level two years. Finally, the patient maintained his walking capacity and independence for 2 years, before the decline due to the disease. After receiving a two-lung transplant, the patient remained intubated for 12 h, left the intensive care unit after 3.5 days and was discharged after 18 days (average values: 48 h, 7–10 days and 25–35 days, respectively). These results show that systematic and supervised combined training can be safety applied in an IPF patient to maintain functionality and quality of life. In addition, we show that RT can be maintained for as long as necessary without complications.

## 1. Introduction

Idiopathic pulmonary fibrosis (IPF) is a chronic interstitial lung disease that causes dyspnea, exercise intolerance, and hypoxemia. It severely affects patient quality of life (QoL) and has a very poor prognosis, with a median survival after diagnosis of between 3 and 5 years and a 5-years survival of 20–40% [1] or 56% in the last study in Spain [2]. Its prevalence is 13/100,000 in women and 20/100,000 in men. As the disease progresses, there is a progressive and irreversible decline in lung function, affecting lung mechanics (i.e., compliance impairment and volume decreases) and gas exchange (i.e., a reduction in diffusion capacity, increasing dead space and arterial hypoxemia) [3]. These pathophysiological characteristics are exacerbated during exercise, producing abnormally high levels of exertional dyspnea [4].

Given this, pulmonary rehabilitation (PR) plays an important role in IPF, as in all chronic pathologies. Although the evidence level is very high in chronic obstructive pulmonary disease (COPD) [5], the evidence level for PR programs in IPF are not as high [6], and the protocols remain unstandardized. In fact, the 2015 consensus of the American Thoracic Society/European Respiratory Society (ATS/ERS) recognized that PR is severely underused worldwide and showed that interventions of at least 8 weeks are needed to achieve effects on exercise capacity and QoL [7]. Prior to 2015, the number of studies with interventions longer than 8 weeks was very low. In 2013, Kenn et al. [8] identified only two studies (10- and 12-week interventions) from a total of eight studies on PR programs in IPF patients with different levels of severity: 63–83% of predicted forced vital capacity (FVC) and 33–63% of predicted lung diffusion capacity (DLCO). In all studies, aerobic training was prescribed in a basic way as either constant walking at 70–80% of the maximum speed achieved during a six-minute walking test (6MWT) or cycling at 60–80% of maximum heart rate (HRmax). In none of the studies was a cardiopulmonary exercise test (CPET) performed to assess the severity of the disease, to objectively quantify training adaptations (i.e., maximum oxygen uptake—VO2max, maximum ventilation—VEmax, etc.) or to prescribe training sessions (i.e., %VO_2_max). Resistance training, which was not completely well described in some of the studies, consisted of “light exercises” or 1–3 sets of 10–30 reps using elastic bands, light weights, functional exercises with the patient’s body weight, or machines (low loads in kg) [8]. This lack of information about the training load (kg or perceived intensity) and even the exercises used (e.g., sometimes they are just described as lower-upper limb exercises) makes it difficult to apply the results in an individualized training program for an IPF patient. 

In a recent meta-analysis [9], seven randomized controlled trials (RCTs) in which the intervention duration was increased to 10–12 weeks (some of them with a follow-up of 3–30 months) were carried out to assess the effects of PR programs on exercise capacity, QoL and lung function deterioration in IPF patients. The results of these studies show positive short-term effects from PR on exercise capacity (6MWT) and lung function (FVC, DLCO and respiratory perception questionnaires).

However, there is a lack of information about the long-term effects of these interventions. Vainshelboim et al. evaluated the short- and long-term effects of a 12-week supervised PR program [10,11]. Although positive short-term effects were found after the intervention [11], at the 11- and 30-month follow-up, there were no differences (vs control group) in functional variables (6MWT, FVC or DLCO) or survival [11]. This clearly suggests the need to prolong supervised training to avoid losing the adaptations induced. Finally, although none of the studies have reported negative events and all of the studies have shown that PR is safe in IPF patients [12], we do not know of any study that has carried out a PR program indefinitely in IPF patients to maintain, or even improve, these positive and vital adaptations.

When measuring these training adaptations, it is surprising that only one study (Vainshelboim 2018 [13]) performed a CPET, just to compare it with the 6MWT and not to quantify its magnitude by ergospirometric variables such as VO_2_max, VEmax, ventilator equivalents of oxygen (VE/VO_2_) and carbon dioxide (VE/VCO2) at the 1st ventilatory threshold (VT1), etc. These values not only quantify adaptations but also have prognostic value for the evolution of the disease [14]. Functional evaluation (through a CPET) for training prescription and control is common in other pulmonary diseases, such as pulmonary arterial hypertension [15] or COPD [16]. 

Perhaps this lack of evaluation in IPF patients could be the reason why most of these studies have measured adaptation with functional tests (such as the 6MWT) or even questionnaires [17] and why none of them describe in detail the prescription and control of the training load (e.g., endurance training intensity expressed in %VO_2_max, resistance exercises and load expressed in kg, etc.). Given this, in 2016, Vainshelboim et al. published the first guidelines for IPF patients [18], which recommended the inclusion of medium-intensity aerobic interval training (for the first time); however, resistance training was not clearly defined in relation to the determination of loads and load control, making it difficult to apply the guidelines.

This is especially important given that the decrease in exercise resulting from dyspnea provokes a progressive loss of muscle mass, which increases fatigue and therefore reinforces inactivity and impairs functional independence. This is very dangerous in a disease characterized by an irreversible and degenerative decline in lung function given the following: (1) the degree of functional independence of the patient before transplantation is related to long-term survival [19,20] and (2) this functional independence is difficult to maintain over a long period with short interventions [10]. Furthermore, it is known that there is a huge decrease in physical capacity after lung transplantation [21,22,23], which is partially due to the loss of muscle mass [24]. Therefore, stopping this inertial deterioration by applying structured and supervised resistance training seems crucial.

One noteworthy aspect of PR interventions in IPF patients is that, although most of the PR studies include breathing retraining (control techniques aimed at reducing respiratory frequency), relaxation techniques, flexibility modes and deep breathing exercises [8,9], only one intervention has included inspiratory muscle training [25]. This therapy is important given its positive long-term effects (one-year training) in pulmonary diseases such as COPD [26] and when it is combined with skeletal muscle training for other pathologies such as cystic fibrosis [27] and pulmonary arterial hypertension [28,29].

The aim of this work was to test the efficacy of a supervised combined (aerobic + resistance + inspiratory) training program to improve positive adaptations and maintain functional independence in an IPF patient until lung transplantation. This program was applied indefinitely, and it is characterized by (1) aerobic training, combining constant and high-intensity interval training, (2) high-load resistance training and (3) inspiratory muscle training, performed with a conventional pressure threshold device.

## 2. Materials and Methods 

The patient was a sedentary 56-year-old man (height: 168 cm, weight: 84.4 kg, BMI: 29.8 kg·m^−2^) diagnosed with IPF on 27 October 2016, in the Interstitial Pneumopathies Unit of Virgen del Rocío Hospital (Seville, Spain). The patient had no other pathology and was not undergoing any type of medical treatment. He was classified as moderately affected due his lung function test: FVC: 2.42 L (62.5% of predicted value.) and DCLO: 37.5% pred. He started treatment with pirfenidone, reaching a maintenance dose of 800 mg three times a day by the third week. This treatment was maintained throughout the study period. The patient’s follow-up was from diagnosis until 6 March 2020, the date on which he was discharged from Reina Sofía Hospital (Córdoba, Spain) after receiving a two-lung transplant on 17 February 2020.

The patient was informed in detail about the procedure to be followed and gave his written consent, agreeing to follow all the instructions regarding the training program.

The pulmonology department of the patient’s reference hospital was informed about the intervention that was proposed to be carried out with him and, since he had not been prescribed respiratory rehabilitation, they gave their approval to carry it out.

The lung transplant unit to which the patient was assigned, was always informed of the evaluations and the functional status of the patient.

The University Research Ethics Committee was informed about the follow-up to be carried out on this patient and did not object.

### 2.1. Laboratory Tests

His first visit to the Exercise Physiology Laboratory of the Universidad Pablo de Olavide (Seville, Spain) was on 28 November 2016, just one month after diagnosis. After completing a medical history and performing a conventional physical evaluation (as he had already been extensively studied in the hospital), a resting electrocardiogram (ECG) (08SD, BTL Medical Technologies, Praha, Czech Republic) was obtained, and forced spirometry for FVC (CPX ultima, Medical Graphics Corporation, St Paul, NM, USA) and maximal inspiratory pressure (MIP) (RPM, Micro Medical Inc., Chatham, Kent, UK) were measured. Finally, the patient performed a CPET on a cycle ergometer (Ergometrics 900, Ergoline, Barcelona, Spain) with an incremental protocol-modified from that used by Lopes et al. [30] in thoracic sarcoidosis, in which HR (in bpm), ECG and oxygen saturation SpO2 (%) (NONIN 7500, Nonin Medical Inc., Plymouth, MN, USA) were continuously monitored. Following the collection of 2 min of resting data, the patient pedaled for 2 min at 60–65 rpm without resistance, after which the work rate (WR) was incrementally increased by 1 W every 6 s (average 10 W/min) until exhaustion or until SpO_2_ dropped below 90%. After that, the protocol included a recovery phase, in which the patient pedaled with a reduced pedal cadence for 1 min without resistance, and a 4-min rest period to prevent post-exercise desaturation. A breath-by-breath automatic system (CPX Ultima, Medical Graphics Corporation, St Paul, NM, USA) was used to measure the following gas exchange and respiratory pattern parameters: PeakVO_2_, VE, VE/VO_2_, VE/VCO_2_, tidal volume (VT), inspiratory time (Ti), mean inspiratory flow (VT/Ti), total time (Ttot) and the ratio between inspiratory and total time (Ti/Ttot). Data were collected in 20-s intervals.

During follow-up, a peak WR < 62 watts and a Peak VO_2_ < 13.8 mL/kg/min were considered indicators of a poor prognosis [13]. Ventilatory efficiency, characterized by delta CO_2_ (delta VE/delta VCO_2_) from the beginning of exercise to the 2nd ventilatory threshold (VT2), was used. DeltaCO_2_ values during exercise vary from 19 to 32 in healthy subjects [31], and values exceeding 34 are considered indicative of the inefficiency of the respiratory system [32].

### 2.2. Training Program

According to the results obtained in the evaluation, a combined (aerobic + resistance + inspiratory) training program was initially prescribed:

The aerobic training included two types of exercises. First, interval training (IT) was carried out twice a week to improve aerobic capacity and reduce dyspnea during exercise [33]. After a warm-up involving pedaling for 5 min without a load, the patient pedaled for 10 min at 30 W and 60 rpm and then performed 6 sets of 15 s at 60 W and 100 rpm, separated by 45 s periods at the initial 30 W and 60 rpm. The IT session ended with 6 min of pedaling at 15 W and 60 rpm, and during the last two minutes, the WR and cadence progressively decreased to 0, to avoid any post-exercise SpO_2_ drop. This IT combination allowed the patient to achieve the highest training intensity (% HRmax and % Peak VO_2_) and therefore to achieve the associated positive cardiac and respiratory adaptations, while keeping SpO_2_ above 90%. Then, to increase the aerobic training volume, the remaining five days of the week, the patient was instructed to walk on flat land for one hour at a speed that would allow him to keep his SpO_2_ above 93%.

Resistance training was initially performed twice a week using a low number of reps per set. The training load was at a perceived exertion (RPE) of between 3 and 5 (without pain) and the lowest SpO_2_ being higher than 90% (this was achieved 10–25 s after each set, depending on the exercise). This combination allowed us to increase the load (kg) progressively on each machine. During the first month, the patient performed 2 sets of 6 reps, and he worked in a circuit structure on the following machines: leg press (30 kg), chest press (15 kg), back row (7 kg), abdominal (10 kg), and lat pull down (25 kg). After the first month, the number of sets was increased to 3 for all exercises (except abdominal exercises, in which the patient progressively increased to 10 because the RPE was very low without SpO_2_ drops), and the loads (kg) were progressively increased while maintaining an RPE of 3–5 and SpO_2_ > 90%).

Inspiratory muscle training (IMT) was performed using a conventional pressure threshold device (Powerbreathe^®^ Sport Medium Resistance, Powerbreathe Spain—Biocorp Europa; Andoain, Spain) at 50% of MIP (initially 60 cm H_2_O). The patient started with 20 maximum inspirations (increasing progressively to 30) once a day during the first month. Inspirations were performed in sets of 5 to avoid drops in SpO_2_ (that occurred after the 6th-8th consecutive inspiration). After the first month, the patient was able to perform all the inspirations consecutively (without SpO_2_ drops), so IMT was performed twice a day (starting with 20 inspirations and increasing to 30 during the second month). From that moment, IMT was kept at 30 inspirations, twice a day, 7 days/week, at 50% of MIP (with MIP being reevaluated during each visit to the lab).

All aerobic, resistance and IMT routines were established by successive tests throughout the first four weeks. During the first four weeks, all the work was monitored and supervised by the researchers. The rest of the time the patient worked autonomously, keeping a work diary in which, he recorded all incidents, including SpO_2_ values (always > 90% as the security criteria) as well as his daily weight. The patient was evaluated in the laboratory every 3 months, using the entire battery of functional assessment tests carried out during the first evaluation. The hospital was informed of each evaluation by reports. Loads were controlled and adjusted weekly according to the diary and laboratory results until training could not be performed with SpO_2_ > 90%.

## 3. Results

The results of the training with body weight are shown in Figure 1c. In the transplant unit, the patient was given the target to reduce his body weight below 75 kg. This goal was reached after 8 months of training and was maintained until the middle of the second year, in which the patient himself set a new goal of 70 kg.

The patient’s lung function remained clinically stable for the first two years of training (Figure 1). However, there was a decrease in DLCO that started in December 2018, the month in which the patient was included on the transplant list because he began to need supplemental oxygen since his SpO_2_ decreased when walking. His DLCO reached 29% shortly before lung transplantation (Figure 1d).

Despite the clinical deterioration, FVC of the patient remained close to that at the time of diagnosis throughout the training period (Figure 1a), and his MIP showed an increase throughout the first year until reaching values of 200 cm H_2_O (Figure 1b). The initial MIP value was 125 cm H_2_O, that is, 110% of the theoretical value calculated with Black and Hyatt equation [34]. At the end of the 2nd year of training (October 2018), IMT was suspended due to the patient’s persistent cough. However, the MIP values remained stable at high values until the moment of the transplant.

The laboratory ergospirometric data evolution during follow-up is shown in Table 1. In the first visit, the maximal power reached was 95 W.

Figure 2 shows the Peak VO_2_ evolution. It increased during the first 1.5 years of training and then decreased from July 2018. Since February 2019, the patient was considered to have a poor prognosis [14].

The respiratory pattern data are shown in Figure 3. VEmax and VT/Ti increased during the first year of training before progressively decreasing until lung transplantation. At the same time, delta CO_2_ was maintained (and even slightly improved) until January 2018, before subsequently deteriorating. However, the Ti/Ttot ratio remained constant throughout the whole follow-up.

IT on the cycle ergometer was maintained until August 2018, at which point it had to be suspended due to SpO_2_ drops below 90%. Table 2 shows the walking training load during the follow-up.

Figure 4 shows that the ability to walk was maintained for the first two years and decreased progressively during the third year until the lung transplant.

Table 3 shows the evolution of the resistance session training loads for each machine. Despite the decrease in tolerance throughout 2019, only the leg press had to be stopped due to post-exercise SpO_2_ < 90% with very slow recovery. We opted to include butterfly work to encourage more trunk hypertrophy to better prepare the patient to tolerate transplant surgery. During this period, the patient needed to train with supplemental oxygen.

Figure 5 shows the total resistance training load expressed as the sum (kg) of all repetitions of all exercises in each training session. This total load increased until 2018 (reaching a plateau during the second half of the year) with a subsequent decrease that ended in a final stabilization phase, which was maintained until the lung transplant.

## 4. Discussion

The main finding of our study was that combined (aerobic + resistance + inspiratory) training, applied for three years (until lung transplantation) in an IPF patient, allowed the patient to maintain walking capacity (without SpO_2_ drops), functional independence and resistance training tolerance for two years before experiencing a decline in functional capacity due to disease. There were several adaptations to training that, in our opinion, are related to those two years of functional maintenance.

It seems clear that adaptations to IMT were the fastest and most supportive of other adaptations. In fact, the MIP increased from 125 to 180 cm H_2_O in the first six weeks and stayed above that value even during the final deterioration phase before the transplant (Figure 1). This improvement could be involved in the fact that the FVC of this patient was maintained for three years, despite the deterioration in other functional variables.

This MIP increase was also described in the only short term IMT study in IPF [24], and it should have contributed, at least in part, to increase the capacity to accumulate aerobic training in different ways. On the one hand, it is known that IMT produces; 1) an increase in inspiratory muscle economy, reducing the oxygen cost of the respiratory muscles and therefore whole body cost in exercise [35], and; 2) a decrease in fatigue, which reduces the fatigue-induced respiratory metabolic reflex, which reduces blood flow to locomotor muscles [36]. These two adaptations jointly facilitate the ability to walk at a certain intensity for a long time without decreasing the SpO_2_. On the other hand, it is also known that IMT improves the strength of the inspiratory musculature, allowing lung pathology patients to overcome and “train” pulmonary compliance [35,37]. This allows patients to maintain a larger alveolar gas exchange surface, as there is a better expansion of lungs, and to maintain a higher VE at a high intensity of exercise. Both effects can help to delay the SpO_2_ drop at high intensity of exercise by several seconds. Therefore, it could help to tolerate longer high-intensity bouts during IT.

Resistance training (except leg press) could be maintained without desaturation until lung transplantation, by adjusting loads depending on the RPE and SpO_2_ and increasing recovery times between sets. This caused an average duration of 2 h per training session in 2019. Although the load decreased, it remained higher than that at the beginning of the follow-up. This must have played an important role in maintaining functional independence and aerobic training capacity. It is known that increases in maximum strength decrease the percentage of maximum strength demanded by daily life activities (i.e., climbing stairs, carrying bags, etc.), thereby allowing the same absolute submaximal load actions (kg or intensity) to be maintained with less oxygen cost [38]. This reduction in oxygen cost of the active musculature in submaximal actions, therefore, stresses the limited diffusion capacity of IPF patients less during daily life activities. Although its magnitude was not measured, which is one of the limitations of this study, the increase in maximum strength of the patient also facilitated tolerance of blocks of IT.

Another reason why we maintained strength training, even when having to do it with supplemental oxygen during the third year, was to prepare the patient for lung transplantation. It is known that the maintenance of muscle strength and independence during the transplant waiting period is an essential factor in the subsequent prognosis [19,20,24]. However, in the literature, PR interventions are short-term, even though the effects of long-term training are known [9]. Therefore, it does not seem very coherent that permanent interventions have not been considered to maintain the physical condition of the patients, especially their muscular strength.

Supported by IMT and strength training adaptations, aerobic training, especially IT, seems to be responsible for the increase in Peak VO_2_ and other ergospirometric parameters measured via CPET during the first year of training and, just as important, for keeping them above the poor prognosis level for almost two years (Figure 2 and Figure 3). For this reason, functional independence was maintained at an acceptable level during these years. During the third year, the progressive decrease in DLCO (Figure 1) (as the logical evolution of the disease would indicate) decreased the tolerance to training (Table 2, Figure 4 and Figure 5) and caused an inability to maintain IT. During such exercise, the difference between the oxygen consumed in the muscles and the oxygen diffused in the lungs became too large. However, we believe that, until that moment, IT had been compensating in some way and slowing down functional impairment.

From the perspective of exercise physiology, this “whole muscle training” approach is analogous to sport training. We need to establish a training program to maintain the best possible condition until the “competitive event”, which, in this case, would be lung transplantation, without sarcopenia. Achieving this goal is key to reducing possible post-surgery-related complications [24] and improving recovery, given that postoperative mortality increases with mechanical ventilation time (>48 h) and with the length of stay in the intense care unit (ICU) (>7 days) [2].

As far as we know, no interventions of this nature have been previously published. The patient received a two-lung transplant on February 17, 2020. He remained intubated for only 12 h, left the ICU after 3.5 days and was discharged after 18 days.

Even though we are aware of the limitations of a case study, in which the results must be interpreted with caution and cannot be generalized to other patients, the postoperative evolution of our patient was surprising. In the bibliography, the average intubation time was 48 h (range 36–144) [38]; the average ICU stay was between 7 days (range 5–19) [39] and 10 days (range 7–23) [38]; and the average stay in the hospital was between 25 (range 20–39) [39] and 34.5 (range 26.5–59.5) days [38].

Undoubtedly, the patient’s good evolution could have been influenced by other factors related to his own physical nature or to any other situation, but in light of the scientific information available [10,11,13,18,19,20,24], we believe that the physical condition maintained by the subject throughout the period from diagnosis to transplantation was, at least in part, a determining factor.

## 5. Conclusions

The results of this study show that a systematic and supervised combined (aerobic + resistance + inspiratory) training can maintain functionality and quality of life in an IPF patient. Additionally, strength training can be maintained for as long as necessary without causing SpO2 drops. These results also suggest that, in the event of needing a lung transplant, maintaining muscle strength during the presurgical period can be key in the patient’s recovery during the immediate postsurgical period.

## Figures and Tables

**Figure 1 ijerph-17-05091-f001:**
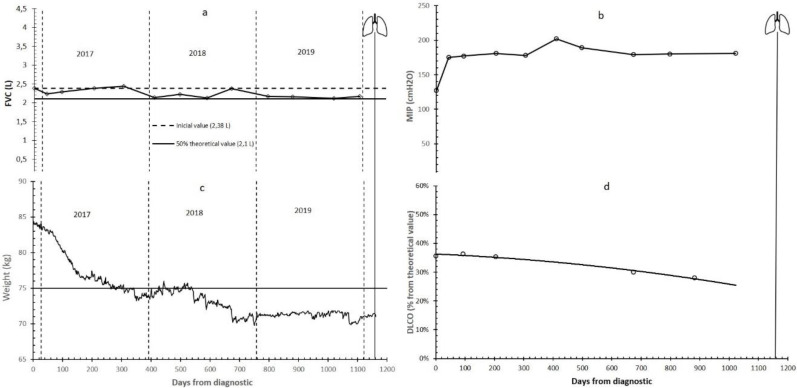
Weight and lung function parameters evolution. (**a**) FVC: forced vital capacity (L); (**b**) MIP: maximum inspiratory pressure (cm H2O); (**c**) body weight (kg); (**d**) DLCO: diffusion lung capacity of carbon monoxide (%). In panel a, the dashed line represents the initial FVC value and the continuous line represents 50% of its theoretical value (2.1 L). In panel c, the horizontal line represents the weigh required by hospital for lung transplantation. Lungs symbol: bilateral pulmonary transplant.

**Figure 2 ijerph-17-05091-f002:**
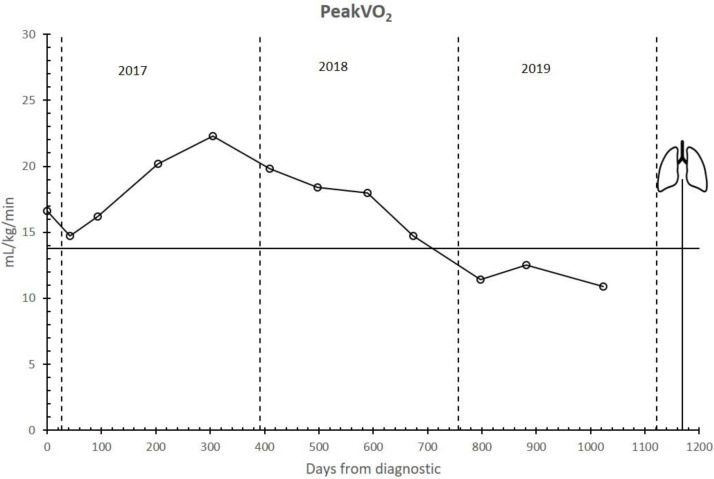
PeakVO_2_ evolution. VO_2_: oxygen uptake (mL·kg^−1^·min^−1^). Lungs symbol: bilateral pulmonary transplant. The horizontal line represents the level of poor prognosis.

**Figure 3 ijerph-17-05091-f003:**
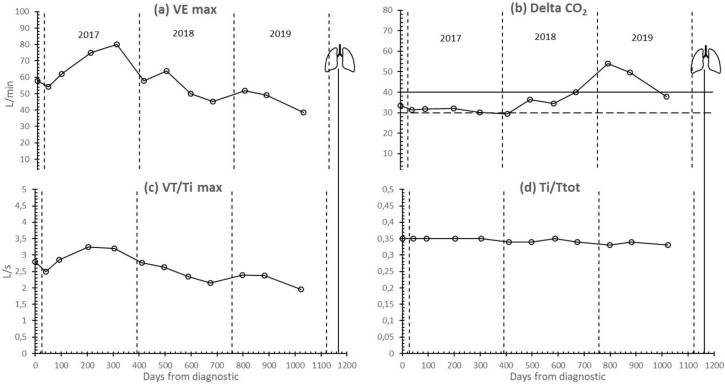
Respiratory pattern evolution. (**a**) VE: ventilation (L·min-1); (**b**) Delta CO_2_; ventilatory efficiency; (**c**) VT·Ti-1: mean inspiratory flow (mL·sec-1); (**d**) Ti·Ttot-1: ratio between inspiratory and total time. Lungs symbol: bilateral pulmonary transplant.

**Figure 4 ijerph-17-05091-f004:**
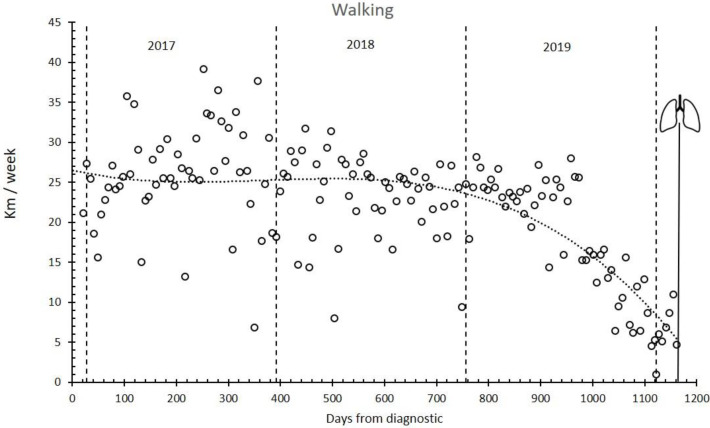
Walking distance. Tolerance to walk (km·week-1) evolution during follow-up. Lungs symbol: bilateral pulmonary transplant.

**Figure 5 ijerph-17-05091-f005:**
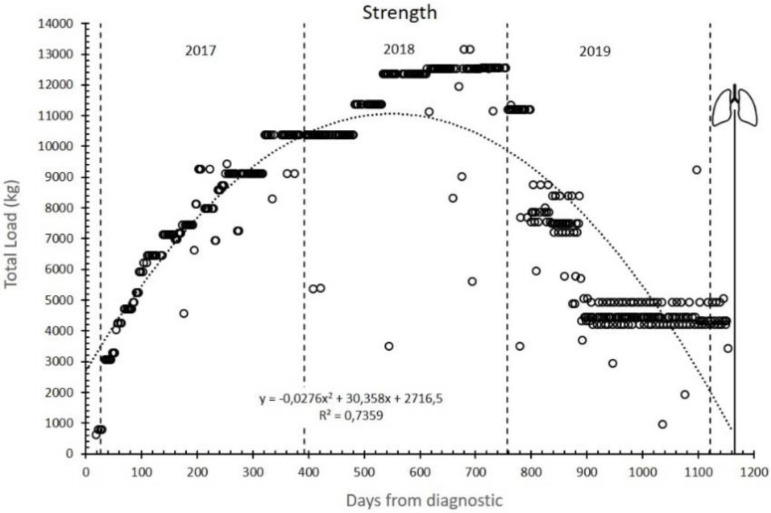
Strength loads: total resistance training load (kg) during the follow-up. Lungs symbol: bilateral pulmonary transplant.

**Table 1 ijerph-17-05091-t001:** Effort Respiratory Data.

	PeakVO_2_	VE/VO_2_	VE/VCO_2_	DeltaCO_2_	VEmax	VT/Ti max	Ti/Ttot
	(mL/kg/min)	(at VT1)	(at VT1)		(L/min)	(L/s)	
28/11/2016	16.6	40	37	33.6	57.9	2.8	0.3
09/01/2017	14.7	31	38	31.3	54.1	2.5	0.3
01/03/2017	16.2	33	36	31.8	61.9	2.8	0.3
20/06/2017	20.2	42	35	32	75	3.2	0.3
29/09/2017	22.3	37	35	30.1	80	3.2	0.3
12/01/2018	19.8	40	34	29.4	57.7	2.7	0.3
09/04/2018	18.4	48	40	36.3	63.9	2.6	0.3
10/07/2018	18.0	37	40	34.5	50.1	2.3	0.3
03/10/2018	14.7	45	46	40	45.3	2.1	0.3
04/02/2019	11.4	61	59	54	51.9	2.4	0.3
29/04/2019	12.5	48	53	49.5	49.2	2.4	0.3
17/09/2019	10.9	50	54	37.9	38.6	1.9	0.3

VO2: oxygen uptake (mL·kg^−1^·min^−1^). VE·VO2-1: ventilatory equivalent of oxygen. VE·VCO2-1; ventilatory equivalent of carbon dioxide. VE: ventilation (L·min^−1^), VT·Ti-1: mean inspiratory flow (mL·sec^−1^), Ti·Ttot-1: ratio between inspiratory and total time).

**Table 2 ijerph-17-05091-t002:** Walking data along the three years follow-up.

		**Kms**	**Min**	**Steps**	**Kcal**
2017	TOTAL	1357	18,497	2,036,787	223,807
	Daily average ± SD	3.7 ± 1.4	51 ± 23	5580 ± 2431	613 ± 293
	Weekly average ± SD	26 ± 5.2	355 ± 78	39,062 ± 6832	4292 ± 822
2018	TOTAL	1221	17,535	1,833,589	180,219
	Daily average ± SD	4.3 ± 1.8	62 ± 25	5024 ± 2653	494 ± 276
	Weekly average ± SD	23.4 ± 4.9	336 ± 73	35,180 ± 7458	3457.6 ± 797
2019	TOTAL	969	14,425	1,455,616	133,821
	Daily average ± SD	2.7 ± 1.1	40 ± 15	3988 ± 1620	367 ± 194
	Weekly average ± SD	18.6 ± 6.9	277 ± 94	27,964 ± 10,438	2572.2 ± 1159

Distance (km), time (min), steps and energetic expenditure (kcal) are expressed as total, daily average and week average. SD: standard deviation.

**Table 3 ijerph-17-05091-t003:** Resistance Training Program.

Exercises	2016 December	2017 June	2017 December	2018 June	2018 December	2019 June	2019 December
Leg Press	3 × 6 × 30 kg (3–94%)	3 × 10 × 61 kg (4–94%)	3 × 10 × 69 kg (4–94%)	3 × 10 × 69 kg (4–94%)	3 × 10 × 70 kg (5–92%)	---	---
Butterfly	---	---	---	---	---	3 × 8 × 30 kg (3–91%)	3 × 8 × 20 kg (5–91%)
Chest press	2 × 6 × 15 kg (3–95%)	3 × 10 × 35 kg (4–94%)	3 × 10 × 35 kg (4–94%)	3 × 10 × 35 kg (4–94%)	3 × 10 × 40 kg (4–92%)	3 × 8 × 30 kg (3–94%)	3 × 8 × 30 kg (5–95%)
Back Row	2 × 6 × 7 kg (4–95%)	3 × 10 × 35 kg (4–94%)	3 × 10 × 35 kg (4–94%)	3 × 10 × 35 kg (4–94%)	3 × 10 × 35 kg (3–93%)	3 × 8 × 30 kg (3–94%)	3 × 8 × 25 kg (3–95%)
Abdominal	2 × 10 × 10 kg (4–94%)	8 × 15 × 30 kg (4–93%)	10 × 20 × 25 kg (4–94%)	10 × 20 × 35 kg (5–93%)	10 × 20 × 35 kg (5–92%)	8 × 15 × 25 kg (6–94%)	8 × 15 × 25 kg (6–96%)
Lat Pull Down	2 × 6 × 25 kg (4–95%)	3 × 10 × 35 kg (4–93%)	3 × 10 × 40 kg (4–93%)	3 × 10 × 40 kg (4–94%)	3 × 10 × 40 kg (5–93%)	3 × 8 × 30 kg (3–92%)	3 × 8 × 30 kg (5–95%)

Data are expressed in series × repetitions × kg (RPE – SpO_2_%). RPE: rating of perceived exertion (0–10), SpO_2_: lowest oxygen saturation (%) after a set. Training sessions needed supplementary oxygen during 2019.

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
