# Peer review of "Long-Term Combined Training in Idiopathic Pulmonary Fibrosis: A Case Study"

_ijerph, 2020, doi:10.3390/ijerph17145091_

Round 1

Reviewer 1 Report

Minimizing postoperative complications is a challenge in which all health professionals must be involved. It is known that a correct preparation before a surgical intervention supposes great benefits for the patient when facing this process. This work provides clinical evidence of the need for specific interventions for these types of patients.

The main disadvantage of this type of work, recognized by the authors themselves, is that the conclusions are based exclusively on observations made on a single individual. The results could be influenced by many variables not studied. The main recommendation is to create a significant sample of patients with similar pathologies so that the results of the intervention are statistically supported.

On the other hand, it is necessary to give more information about the patient included in the study. No reference is made to other pathologies or pharmacological treatments, only mention is made of treatment with pirfenidone.

As minor corrections:

- line 90, page 2: The authors begin to list points related to worsening lung function and transplantation, however, only number 1 appears (1)).

- The format of the tables should be reviewed, since it is not necessary to consider them as figures.

- Table 2, page 7: To complete the data, the dispersion measures associated with the measures of central tendency should be included, such as the standard deviation of the mean.

Best reguards.

Author Response

RESPONSES TO REVIEWER 1

POINT 1.

The main disadvantage of this type of work, recognized by the authors themselves, is that the conclusions are based exclusively on observations made on a single individual. The results could be influenced by many variables not studied. The main recommendation is to create a significant sample of patients with similar pathologies so that the results of the intervention are statistically supported.

Response 1.

Thank you for your comment. Indeed, the limitation of a case report is precisely that it is a single case.

However, the contribution of this study is very important because it highlights that 1) it is feasible to maintain controlled training in a patient of these characteristics throughout the evolution of the disease and 2) that this training is not only not harmful, but also produces notable advantages in the evolution and responses of the patient.

To create a significant sample with this pathology will be the next step.

POINT 2.

On the other hand, it is necessary to give more information about the patient included in the study. No reference is made to other pathologies or pharmacological treatments, only mention is made of treatment with pirfenidone.

Response 2.

The patient had no other pathology and was not undergoing any type of medical treatment. In the Material and Methods section (line 112) this information has been included.

POINT 3.

Line 90, page 2: The authors begin to list points related to worsening lung function and transplantation, however, only number 1 appears (1)).

Response 3.

Thanks for the warning. In line 91 we have added "and 2)".

POINT 4.

The format of the tables should be reviewed, since it is not necessary to consider them as figures.

Response 4.

Excuse me, but I do not understand the comment well. I don't know what exactly you mean by “it is not necessary to consider them as figures”.

However, we have made some changes to the tables suggested by your next point.

POINT 5.

Table 2, page 7: To complete the data, the dispersion measures associated with the measures of central tendency should be included, such as the standard deviation of the mean.

Response 5.

The standard deviation of the mean has been included in Table 2.

Reviewer 2 Report

This manuscript presents the treatment of a single male patient (case study) for the condition IPF. The three-fold training approach warranted impressive results over the first two years and the patient condition progressively deteriorated thereafter. The authors suggest the training improved patient outcomes up to lung transplant. I have a few comments for revision and they are listed below:

  1. What is the prevalence of IPF in adults? 
  2. Lines 37-39: Referring to "the evidence level is very high" and only citing one reference is not suggestive of a lot of evidence. It would be great to add what the general outcomes of PR are and then list the different conditions it has been shown to improve with a reference for each.
  3. Lines 77-78 could be part of the previous paragraph and "CODP" should be "COPD" (line 78).
  4. The lack of available information is highlighted throughout the introduction. It is clear the long-term evidence and follow-up is needed but again leads to point #1, what is the prevalence of the condition? Is there a lack because the patient number is so low? Not that this intervention is not important but potentially why the evidence is lacking.
  5. Table 1 results: the values behind the decimal should be consistent.
  6. Figure 2 and lines 221-222: It appears the VO2 started dropping 300 days after diagnostic.  What does the horizontal line at approximately 13.8 mL/kg/min represent?
  7. The text never calls out Figure 2.
  8. Table 2: The row titles do not match the foot note description (translation error, perhaps?)
  9. The progression of exercise loads is well documented. I believe the discussion is a bit misleading though. There is a visual decline in performance(distance, load, etc.) after about 1.5 years. A day value at which the pt is placed on the transplant list would be a useful reference point. 
  10. It would also be interesting to see the follow-up. It is clear this patient had much shorter ICU time post-op. How well did the patient receive the transplant and can he continue to participate in a modified exercise routine. 
  11. Do the authors think the improved outcomes are due to the physical fitness of the patient? Does/did he have fewer comorbidities than other "typical" IPF patients?
  12. Is there any desire to prolong the follow-up?

Author Response

RESPONSES TO REVIEWER 2

Thank you very much for your comments.

POINT 1.

What is the prevalence of IPF in adults?

Response 1.

Its prevalence is 13 / 100,000 in women and 20 / 100,000 in men. It was included in line 31.

POINT 2.

Lines 37-39: Referring to "the evidence level is very high" and only citing one reference is not suggestive of a lot of evidence. It would be great to add what the general outcomes of PR are and then list the different conditions it has been shown to improve with a reference for each.

Response 2.

The idea was to express that for Chronic Obstructive Pulmonary Disease there is a lot of evidence about the benefits of PR, while this is not the case for IPF. A single article is cited (Puhan, MA; Lareau, SC Evidence-based outcomes from pulmonary rehabilitation in the chronic obstructive pulmonary disease patient. Clinics in Chest Medicine 2014, 35, 295-301) because it includes the existing evidence in the case of COPD.

We act this way in order not to dwell on this aspect since it is not the subject of the paper.

POINT 3.

Lines 77-78 could be part of the previous paragraph and "CODP" should be "COPD" (line 78).

Response 3.

Lines 77-78 have been added to the previous paragraph and "CODP" was changed by "COPD".

POINT 4.

The lack of available information is highlighted throughout the introduction. It is clear the long-term evidence and follow-up is needed but again leads to point #1, what is the prevalence of the condition? Is there a lack because the patient number is so low? Not that this intervention is not important but potentially why the evidence is lacking.

Response 4.

The problema about the lack of information is not the prevalence. To our knowledge, the problem is that PR is not sistematically used in patients with IPF. We state in the text (lines 40-42) that “the 2015 consensus of the American Thoracic Society/European Respiratory Society (ATS/ERS) recognized that PR is severely underused worldwide and showed that interventions of at least 8 weeks are needed to achieve effects on exercise capacity and QoL”.

But this is not the only problem. In our opinión, it is very important to consider that, in addition to being underuse, the PR programs in IPF are short-term and in no case they are systematized to maintain the patient's physical condition with a view to lung transplantation or simply to maintain the best possible quality of life. These aspects are reduced to very few studies, but are not part of the routine.

POINT 5.

Table 1 results: the values behind the decimal should be consistent.

Response 5.

All values in Table 1 have been unified to one decimal.

POINT 6.

Figure 2 and lines 221-222: It appears the VO2 started dropping 300 days after diagnostic.  What does the horizontal line at approximately 13.8 mL/kg/min represent?

Response 6.

The horizontal line represents the level of poor prognosis according to Vainshelboim et al (2018). This observation has been added to the Figure 2 caption.

POINT 7.

The text never calls out Figure 2.

Response 7.

The text on lines 223-225 corresponds to the main text but appears as if it were a figure caption. This has been corrected by marking it in red. This text is not a figure caption; it is main text calling out figure 2.

POINT 8.

Table 2: The row titles do not match the foot note description (translation error, perhaps?).

Response 8.

Sorry! It was a mistake!

The row titles have been changed in table 2 acording to the foot note description.

POINT 9.

The progression of exercise loads is well documented. I believe the discussion is a bit misleading though. There is a visual decline in performance (distance, load, etc.) after about 1.5 years. A day value at which the pt is placed on the transplant list would be a useful reference point.

Response 9.

The inclusion of the patient on the transplant waiting list occurred on November 13, 2018. This corresponds to day 708 of follow-up. Being so close to the end of 2018, the vertical line that marks the change from 2018 to 2019 serves as a reference.

In lines 200-202 we say: “The patient’s lung function remained clinically stable for the first two years of training (figure 1). However, there was a decrease in DLCO that started in December 2018, the month in which the patient was included on the transplant list because he began to need supplemental oxygen”.

POINT 10.

It would also be interesting to see the follow-up. It is clear this patient had much shorter ICU time post-op. How well did the patient receive the transplant and can he continue to participate in a modified exercise routine.

Response 10.

The patient received the transplant very well and not only the ICU stay, but also all the evolution was surprising. Although it is no longer part of this paper (because it ends with the transplant), we have continued to monitor the patient and I can tell you that at this time (4 months after discharge) he is training again. At the moment the strength loads are still low, but the inspiratory muscle training is above what he did at the beginning of the follow-up.

POINT 11.

Do the authors think the improved outcomes are due to the physical fitness of the patient? Does/did he have fewer comorbidities than other "typical" IPF patients?

Response 11.

We are convinced that the good post-surgery evolution of the patient has a lot to do with his physical fitness and we strongly recomend this method in all IPF patients, both waiting transplant or not.

POINT 12.

Is there any desire to prolong the follow-up?

Response 12.

As said in response 10, we have continued the patient follow up and training program.